# Learning ReLUs via Gradient Descent

**Mahdi Soltanolkotabi**
Ming Hsieh Department of Electrical Engineering
University of Southern California
Los Angeles, CA
soltanol@usc.edu

## Abstract

In this paper we study the problem of learning Rectified Linear Units (ReLUs) which are functions of the form $x \mapsto \max(0, \langle w, x \rangle)$ with $w \in \mathbb{R}^d$ denoting the weight vector. We study this problem in the high-dimensional regime where the number of observations are fewer than the dimension of the weight vector. We assume that the weight vector belongs to some closed set (convex or nonconvex) which captures known side-information about its structure. We focus on the realizable model where the inputs are chosen i.i.d. from a Gaussian distribution and the labels are generated according to a planted weight vector. We show that projected gradient descent, when initialized at $\mathbf{0}$, converges at a linear rate to the planted model with a number of samples that is optimal up to numerical constants. Our results on the dynamics of convergence of these very shallow neural nets may provide some insights towards understanding the dynamics of deeper architectures.

## 1   Introduction

Nonlinear data-fitting problems are fundamental to many supervised learning tasks in signal processing and machine learning. Given training data consisting of $n$ pairs of input features $x_i \in \mathbb{R}^d$ and desired outputs $y_i \in \mathbb{R}$ we wish to infer a function that best explains the training data. In this paper we focus on fitting Rectified Linear Units (ReLUs) to the data which are functions $\phi_w : \mathbb{R}^d \to \mathbb{R}$ of the form

$$\phi_{\boldsymbol{w}}(\boldsymbol{x}) = \max\left(0, \langle \boldsymbol{w}, \boldsymbol{x} \rangle\right).$$

A natural approach to fitting ReLUs to data is via minimizing the least-squares misfit aggregated over the data. This optimization problem takes the form

$$\min_{\boldsymbol{w} \in \mathbb{R}^d} \quad \mathcal{L}(\boldsymbol{w}) := \frac{1}{n} \sum_{i=1}^{n} \left(\max\left(0, \langle \boldsymbol{w}, \boldsymbol{x}_i \rangle\right) - y_i\right)^2 \quad \text{subject to} \quad \mathcal{R}(\boldsymbol{w}) \le R, \qquad (1.1)$$

with $\mathcal{R} : \mathbb{R}^d \to \mathbb{R}$ denoting a regularization function that encodes prior information on the weight vector.

Fitting nonlinear models such as ReLUs have a rich history in statistics and learning theory [12] with interesting new developments emerging [6] (we shall discuss all these results in greater detail in Section 5). Most recently, nonlinear data fitting problems in the form of neural networks (a.k.a. deep learning) have emerged as powerful tools for automatically extracting interpretable and actionable information from raw forms of data, leading to striking breakthroughs in a multitude of applications [13, 15, 4]. In these and many other empirical domains it is common to use local search heuristics such as gradient or stochastic gradient descent for nonlinear data fitting. These local search heuristics are surprisingly effective on real or randomly generated data. However, despite their empirical success the reasons for their effectiveness remains mysterious.

Focusing on fitting ReLUs, a-priori it is completely unclear why local search heuristics such as gradient descent should converge for problems of the form (1.1), as not only the regularization function maybe nonconvex but also the loss function! Efficient fitting of ReLUs in this high-dimensional setting poses new challenges: When are the iterates able to escape local optima and saddle points and converge to global optima? How many samples do we need? How does the number of samples depend on the a-priori prior knowledge available about the weights? What regularizer is best suited to utilizing a particular form of prior knowledge? How many passes (or iterations) of the algorithm is required to get to an accurate solution? At the heart of answering these questions is the ability to predict convergence behavior/rate of (non)convex constrained optimization algorithms. In this paper we build up on a new framework developed in the context of phase retrieval [21] for analyzing nonconvex optimization problems to address such challenges.

## 2 Precise measures for statistical resources

We wish to characterize the rates of convergence for the projected gradient updates (3.2) as a function of the number of samples, the available prior knowledge and the choice of the regularizer. To make these connections precise and quantitative we need a few definitions. Naturally the required number of samples for reliable data fitting depends on how well the regularization function $\mathcal{R}$ can capture the properties of the weight vector $\boldsymbol{w}$. For example, if we know that the weight vector is approximately sparse, naturally using an $\ell_1$ norm for the regularizer is superior to using an $\ell_2$ regularizer. To quantify this capability we first need a couple of standard definitions which we adapt from [17, 18, 21].

**Definition 2.1 (Descent set and cone)** *The* set of descent *of a function $\mathcal{R}$ at a point $\boldsymbol{w}^*$ is defined as*

$$\mathcal{D}_{\mathcal{R}}(\boldsymbol{w}^*) = \left\{ \boldsymbol{h} : \ \mathcal{R}(\boldsymbol{w}^* + \boldsymbol{h}) \le \mathcal{R}(\boldsymbol{w}^*) \right\}.$$

*The* cone of descent *is defined as a closed cone $\mathcal{C}_{\mathcal{R}}(\boldsymbol{w}^*)$ that contains the descent set, i.e. $\mathcal{D}_{\mathcal{R}}(\boldsymbol{w}^*) \subset \mathcal{C}_{\mathcal{R}}(\boldsymbol{w}^*)$. The* tangent cone *is the conic hull of the descent set. That is, the smallest closed cone $\mathcal{C}_{\mathcal{R}}(\boldsymbol{w}^*)$ obeying $\mathcal{D}_{\mathcal{R}}(\boldsymbol{w}^*) \subset \mathcal{C}_{\mathcal{R}}(\boldsymbol{w}^*)$.*

We note that the capability of the regularizer $\mathcal{R}$ in capturing the properties of the unknown weight vector $\boldsymbol{w}^*$ depends on the size of the descent cone $\mathcal{C}_{\mathcal{R}}(\boldsymbol{w}^*)$. The smaller this cone is the more suited the function $\mathcal{R}$ is at capturing the properties of $\boldsymbol{w}^*$. To quantify the size of this set we shall use the notion of mean width.

**Definition 2.2 (Gaussian width)** *The Gaussian width of a set $\mathcal{C} \in \mathbb{R}^d$ is defined as:*

$$\omega(\mathcal{C}) := \mathbb{E}_{\boldsymbol{g}}\big[\sup_{\boldsymbol{z} \in \mathcal{C}} \langle \boldsymbol{g}, \boldsymbol{z} \rangle\big],$$

*where the expectation is taken over $\boldsymbol{g} \sim \mathcal{N}(\boldsymbol{0}, \boldsymbol{I}_p)$. Throughout we use $\mathcal{B}^d/\mathbb{S}^{d-1}$ to denote the the unit ball/sphere of $\mathbb{R}^d$.*

We now have all the definitions in place to quantify the capability of the function $\mathcal{R}$ in capturing the properties of the unknown parameter $\boldsymbol{w}^*$. This naturally leads us to the definition of the minimum required number of samples.

**Definition 2.3 (minimal number of samples)** *Let $\mathcal{C}_{\mathcal{R}}(\boldsymbol{w}^*)$ be a cone of descent of $\mathcal{R}$ at $\boldsymbol{w}^*$. We define the minimal sample function as*

$$\mathcal{M}(\mathcal{R}, \boldsymbol{w}^*) = \omega^2(\mathcal{C}_{\mathcal{R}}(\boldsymbol{w}^*) \cap \mathcal{B}^d).$$

*We shall often use the short hand $n_0 = \mathcal{M}(\mathcal{R}, \boldsymbol{w}^*)$ with the dependence on $\mathcal{R}, \boldsymbol{w}^*$ implied.*

We note that $n_0$ is exactly the minimum number of samples required for structured signal recovery from linear measurements when using convex regularizers [3, 1]. Specifically, the optimization problem

$$\sum_{i=1}^{n} \left(y_r - \langle \boldsymbol{x}_i, \boldsymbol{w}^* \rangle\right)^2 \quad \text{subject to} \quad \mathcal{R}(\boldsymbol{w}) \le \mathcal{R}(\boldsymbol{w}^*), \tag{2.1}$$

succeeds at recovering an unknown weight vector $\boldsymbol{w}^*$ with high probability from $n$ observations of the form $\boldsymbol{y}_i = \langle \boldsymbol{a}_i, \boldsymbol{w}^* \rangle$ if and only if $n \geq n_0$.[1] While this result is only known to be true for convex regularization functions we believe that $n_0$ also characterizes the minimal number of samples even for nonconvex regularizers in (2.1). See [17] for some results in the nonconvex case as well as the role this quantity plays in the computational complexity of projected gradient schemes for linear inverse problems. Given that with nonlinear samples we have less information (we loose some information compared to linear observations) we can not hope to recover the weight vector from $n \leq n_0$ when using (1.1). Therefore, we can use $n_0$ as a lower-bound on the minimum number of observations required for projected gradient descent iterations (3.2) to succeed at finding the right model.

# 3 Theoretical results for learning ReLUs

A simple heuristic for optimizing (1.1) is to use gradient descent. One challenging aspect of the above loss function is that it is not differentiable and it is not clear how to run projected gradient descent. However, this does not pose a fundamental challenge as the loss function is differentiable except for isolated points and we can use the notion of generalized gradients to define the gradient at a non-differentiable point as one of the limit points of the gradient in a local neighborhood of the non-differentiable point. For the loss in (1.1) the generalized gradient takes the form

$$\nabla \mathcal{L}(\boldsymbol{w}) := \frac{1}{n} \sum_{i=1}^{n} \left( \text{ReLU} \left( \langle \boldsymbol{w}, \boldsymbol{x}_i \rangle \right) - y_i \right) \left( 1 + \text{sgn}(\langle \boldsymbol{w}, \boldsymbol{x}_i \rangle) \right) \boldsymbol{x}_i. \tag{3.1}$$

Therefore, projected gradient descent takes the form

$$\boldsymbol{w}_{\tau+1} = \mathcal{P}_{\mathcal{K}} \left( \boldsymbol{w}_\tau - \mu_\tau \nabla \mathcal{L}(\boldsymbol{w}_\tau) \right), \tag{3.2}$$

where $\mu_\tau$ is the step size and $\mathcal{K} = \{ \boldsymbol{w} \in \mathbb{R}^d : \mathcal{R}(\boldsymbol{w}) \leq R \}$ is the constraint set with $\mathcal{P}_{\mathcal{K}}$ denoting the Euclidean projection onto this set.

**Theorem 3.1** *Let $\boldsymbol{w}^* \in \mathbb{R}^d$ be an arbitrary weight vector and $\mathcal{R} : \mathbb{R}^d \to \mathbb{R}$ be a proper function (convex or nonconvex). Suppose the feature vectors $\boldsymbol{x}_i \in \mathbb{R}^d$ are i.i.d. Gaussian random vectors distributed as $\mathcal{N}(\boldsymbol{0}, \boldsymbol{I})$ with the corresponding labels given by*

$$\boldsymbol{y}_i = \max \left( 0, \langle \boldsymbol{x}_i, \boldsymbol{w}^* \rangle \right).$$

*To estimate $\boldsymbol{w}^*$, we start from the initial point $\boldsymbol{w}_0 = \boldsymbol{0}$ and apply the Projected Gradient Descent (PGD) updates of the form*

$$\boldsymbol{w}_{\tau+1} = \mathcal{P}_{\mathcal{K}} \left( \boldsymbol{w}_\tau - \mu_\tau \nabla \mathcal{L}(\boldsymbol{w}_\tau) \right), \tag{3.3}$$

*with $\mathcal{K} := \{ \boldsymbol{w} \in \mathbb{R}^d : \mathcal{R}(\boldsymbol{w}) \leq \mathcal{R}(\boldsymbol{w}^*) \}$ and $\nabla \mathcal{L}$ defined via (3.1). Also set the learning parameter sequence to $\mu_0 = 2$ and $\mu_\tau = 1$ for all $\tau = 1, 2, \ldots$ and let $n_0 = \mathcal{M}(\mathcal{R}, \boldsymbol{w}^*)$, per Definition 2.3, be our lower bound on the number of observations. Also assume*

$$n > c n_0, \tag{3.4}$$

*holds for a fixed numerical constant c. Then there is an event of probability at least $1 - 9e^{-\gamma n}$ such that on this event the updates (3.3) obey*

$$\left\| \boldsymbol{w}_\tau - \boldsymbol{w}^* \right\|_{\ell_2} \leq \left( \frac{1}{2} \right)^\tau \left\| \boldsymbol{w}^* \right\|_{\ell_2}. \tag{3.5}$$

*Here $\gamma$ is a fixed numerical constant.*

The first interesting and perhaps surprising aspect of this result is its generality: it applies not only to convex regularization functions but also nonconvex ones! As we mentioned earlier the optimization problem in (1.1) is not known to be tractable even for convex regularizers. Despite the nonconvexity of both the objective and regularizer, the theorem above shows that with a near minimal number

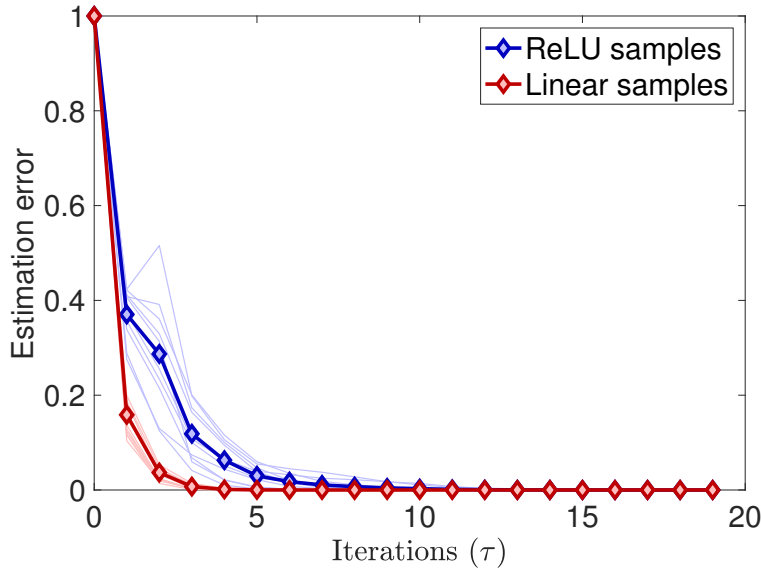

Figure 1: Estimation error ($\|\boldsymbol{w}_\tau - \boldsymbol{w}^*\|_{\ell_2}$) obtained via running PGD iterates as a function of the number of iterations $\tau$. The plots are for two different observations models: 1) ReLU observations of the form $\boldsymbol{y}$ =ReLU($\boldsymbol{X}\boldsymbol{w}^*$) and 2) linear observations of the form $\boldsymbol{y} = \boldsymbol{X}\boldsymbol{w}^*$. The bold colors depict average behavior over 100 trials. None bold color depict the estimation error of some sample trials.

of data samples, projected gradient descent provably learns the original weight vector $\boldsymbol{w}^*$ without getting trapped in any local optima.

Another interesting aspect of the above result is that the convergence rate is linear. Therefore, to achieve a relative error of $\epsilon$ the total number of iterations is on the order of $\mathcal{O}(\log(1/\epsilon))$. Thus the overall computational complexity is on the order of $\mathcal{O}\left(nd\log(1/\epsilon)\right)$ (in general the cost is the total number of iterations multiplied by the cost of applying the feature matrix $\boldsymbol{X}$ and its transpose). As a result, the computational complexity is also now optimal in terms of dependence on the matrix dimensions. Indeed, for a dense matrix even verifying that a good solution has been achieved requires one matrix-vector multiplication which takes $\mathcal{O}(nd)$ time.

## 4 Numerical experiments

In this section we carry out a simple numerical experiment to corroborate our theoretical results. For this purpose we generate a unit norm sparse vector $\boldsymbol{w}^* \in \mathbb{R}^d$ of dimension $d = 1000$ containing $s = d/50$ non-zero entries. We also generate a random feature matrix $\boldsymbol{X} \in \mathbb{R}^{n \times d}$ with $n = \lfloor 8s \log(d/s) \rfloor$ and containing i.i.d. $\mathcal{N}(0,1)$ entries. We now take two sets of observations of size $n$ from $\boldsymbol{\theta}^*$:

- ReLU observations: the response vector is equal to $\boldsymbol{y}$ =ReLU($\boldsymbol{X}\boldsymbol{w}^*$).
- Linear observations: the response is $\boldsymbol{y} = \boldsymbol{X}\boldsymbol{w}^*$.

We apply the projected gradient iterations to both observation models starting from $\boldsymbol{w}_0 = \boldsymbol{0}$. For the ReLU observations we use the step size discussed in Theorem 3.1. For the linear model we apply projected gradient descent updates of the form

$$\boldsymbol{w}_{\tau+1} = \mathcal{P}_\mathcal{K}\left(\boldsymbol{w}_\tau - \frac{1}{n}\boldsymbol{X}^T(\boldsymbol{X}\boldsymbol{w}_\tau - \boldsymbol{y})\right).$$

In both cases we use the regularizer $\mathcal{R}(\boldsymbol{w}) = \|\boldsymbol{w}\|_{\ell_0}$ so that the projection only keeps the top $s$ entries of the vector (a.k.a. iterative hard thresholding). In Figure 1 the resulting estimation errors ($\|\boldsymbol{w}_\tau - \boldsymbol{w}^*\|_{\ell_2}$) is depicted as a function of the number of iterations $\tau$. The bold colors depict average behavior over 100 trials. The estimation error of some sample trials are also depicted in none bold

colors. This plot clearly show that PGD iterates applied to ReLU observations converge quickly to the ground truth. This figure also clearly demonstrates that the behavior of the PGD iterates applied to both models are similar, further corroborating the results of Theorem 3.1. We note that the sample complexity used in this simulation is $8s \log(n/s)$ which is a constant factor away from $n_0 \propto s \log(n/s)$ confirming our assertion that the required sample complexity is a constant factor away from $n_0$ (as predicted by Theorem 3.1).

## 5  Discussions and prior art

There is a large body of work on learning nonlinear models. A particular class of such problems that have been studied are the so called idealized Single Index Models (SIMs) [9, 10]. In these problems the inputs are labeled examples $\{(\boldsymbol{x}_i, y_i)\}_{i=1}^n \in \mathbb{R}^d \times \mathbb{R}$ which are guaranteed to satisfy $y_i = f(\langle \boldsymbol{w}, \boldsymbol{x}_i \rangle)$ for some $\boldsymbol{w} \in \mathbb{R}^d$ and nondecreasing (Lipchitz continuous) $f : \mathbb{R} \to \mathbb{R}$. The goal in this problem is to find a (nearly) accurate such $f$ and $\boldsymbol{w}$. An interesting polynomial-time algorithm called the Isotron exists for this problem [12, 11]. In principle, this approach can also be used to fit ReLUs. However, these results differ from ours in term of both assumptions and results. On the one had, the assumptions are slightly more restrictive as they require bounded features $\boldsymbol{x}_i$, outputs $y_i$ and weights. On the other hand, these result hold for much more general distributions and more general models than the realizable model studied in this paper. These results also do not apply in the high dimensional regime where the number of observations is significantly smaller than the number of parameters (see [5] for some results in this direction). In the realizable case, the Isotron result require $\mathcal{O}(\frac{1}{\epsilon})$ iterations to achieve $\epsilon$ error in objective value. In comparison, our results guarantee convergence to a solution with relative error $\epsilon$ ($\|\boldsymbol{w}_\tau - \boldsymbol{w}^*\|_{\ell_2} / \|\boldsymbol{w}^*\|_{\ell_2} \leq \epsilon$) after $\log(1/\epsilon)$ iterations. Focusing on the specific case of ReLU functions, an interesting recent result [6] shows that reliable learning of ReLUs is possible under very general but bounded distributional assumptions. To achieve an accuracy of $\epsilon$ the algorithm runs in poly($1/\epsilon$) time. In comparison, as mentioned earlier our result rquires $\log(1/\epsilon)$ iterations for reliable parameter estimation. We note however we study the problem in different settings and a direct comparison is not possible between the two results.

We would like to note that there is an interesting growing literature on learning shallow neural networks with a single hidden layer with i.i.d. inputs, and under a realizable model (i.e. the labels are generated from a network with planted weights) [23, 2, 25]. For isotropic Gaussian inputs, [23] shows that with two hidden unites ($k = 2$) there are no critical points for configurations where both weight vectors fall into (or outside) the cone of ground truth weights. With the same assumptions, [2] proves that for a single-hidden ReLU network with a single non-overlapping convolutional filter, all local minimizers of the population loss are global; they also give counter-examples in the overlapping case and prove the problem is NP-hard when inputs are not Gaussian. [25] studies general single-hidden layer networks and shows that a version of gradient descent which uses a fresh batch of samples in each iteration converges to the planted model. This holds using an initialization obtained via a tensor decomposition method. Our approach and convergence results differ from this literature in a variety of different ways. First, we focus on zero hidden layers with a regularization term. Some of this literature focuses on one-hidden layers without (or with specific) regularization. Second, unlike some of these results such as [2, 14], we study the optimization properties of the empirical function, not its expected value. Third, we initialize at zero in lieu of sophisticated initialization schemes. Finally, our framework does not require a fresh batch of samples per new gradient iteration as in [25]. We also note that several publications study the effect of over-parametrization on the training of neural networks without any regularization [19, 8, 16, 22]. Therefore, the global optima are not unique and hence the solutions may not generalize. In comparison we study the problem with an arbitrary regularization which allows for a unique global optima.

## 6  Proofs

### 6.1  Preliminaries

In this section we gather some useful results on concentration of stochastic processes which will be crucial in our proofs. These results are mostly adapted from [21]. We begin with a lemma which is a direct consequence of Gordon's escape from the mesh lemma [7].

**Lemma 6.1** *Assume $\mathcal{C} \subset \mathbb{R}^d$ is a cone and $\mathbb{S}^{d-1}$ is the unit sphere of $\mathbb{R}^d$. Also assume that*

$$n \geq \max\left(20\frac{\omega^2(\mathcal{C} \cap \mathbb{S}^{d-1})}{\delta^2}, \frac{1}{2\delta} - 1\right),$$

*for a fixed numerical constant c. Then for all $\boldsymbol{h} \in \mathcal{C}$*

$$\left|\frac{1}{n}\sum_{i=1}^{n}(\langle \boldsymbol{x}_i, \boldsymbol{h}\rangle)^2 - \|\boldsymbol{h}\|_{\ell_2}^2\right| \leq \delta \|\boldsymbol{h}\|_{\ell_2}^2,$$

*holds with probability at least $1 - 2e^{-\frac{\delta^2}{360}n}$.*

We also need a generalization of the above lemma stated below.

**Lemma 6.2 ([21])** *Assume $\mathcal{C} \subset \mathbb{R}^d$ is a cone (not necessarily convex) and $\mathbb{S}^{d-1}$ is the unit sphere of $\mathbb{R}^d$. Also assume that*

$$n \geq \max\left(80\frac{\omega^2(\mathcal{C} \cap \mathbb{S}^{d-1})}{\delta^2}, \frac{2}{\delta} - 1\right),$$

*for a fixed numerical constant c. Then for all $\boldsymbol{u}, \boldsymbol{h} \in \mathcal{C}$*

$$\left|\frac{1}{n}\sum_{i=1}^{n}\langle \boldsymbol{x}_i, \boldsymbol{u}\rangle\langle \boldsymbol{x}_i, \boldsymbol{h}\rangle - \boldsymbol{u}^*\boldsymbol{h}\right| \leq \delta \|\boldsymbol{u}\|_{\ell_2} \|\boldsymbol{h}\|_{\ell_2},$$

*holds with probability at least $1 - 6e^{-\frac{\delta^2}{1440}n}$.*

We next state a generalization of Gordon's escape through the mesh lemma also from [21].

**Lemma 6.3 ([21])** *Let $\boldsymbol{s} \in \mathbb{R}^d$ be fixed vector with nonzero entries and construct the diagonal matrix $\boldsymbol{S} = diag(\boldsymbol{s})$. Also, let $\boldsymbol{X} \in \mathbb{R}^{n \times d}$ have i.i.d. $\mathcal{N}(0,1)$ entries. Furthermore, assume $\mathcal{T} \subset \mathbb{R}^d$ and define $b_d(\boldsymbol{s}) = \mathbb{E}[\|\boldsymbol{S}\boldsymbol{g}\|_{\ell_2}]$, where $\boldsymbol{g} \in \mathbb{R}^d$ is distributed as $\mathcal{N}(\boldsymbol{0}, \boldsymbol{I}_n)$. Also, define*

$$\sigma(\mathcal{T}) := \max_{\boldsymbol{v} \in \mathcal{T}} \|\boldsymbol{v}\|_{\ell_2}.$$

*Then for all $\boldsymbol{u} \in \mathcal{T}$*

$$\left|\|\boldsymbol{S}\boldsymbol{A}\boldsymbol{u}\|_{\ell_2} - b_d(\boldsymbol{s})\|\boldsymbol{u}\|_{\ell_2}\right| \leq \|\boldsymbol{s}\|_{\ell_\infty}\omega(\mathcal{T}) + \eta,$$

*holds with probability at least $1 - 6e^{-\frac{\eta^2}{8\|\boldsymbol{s}\|_{\ell_\infty}^2 \sigma^2(\mathcal{T})}}$.*

The previous lemma leads to the following Corollary.

**Corollary 6.4** *Let $\boldsymbol{s} \in \mathbb{R}^d$ be fixed vector with nonzero entries and assume $\mathcal{T} \subset \mathcal{B}^d$. Furthermore, assume*

$$\|\boldsymbol{s}\|_{\ell_2}^2 \geq \max\left(20\|\boldsymbol{s}\|_{\ell_\infty}^2 \frac{\omega^2(\mathcal{T})}{\delta^2}, \frac{3}{2\delta} - 1\right).$$

*Then for all $\boldsymbol{u} \in \mathcal{T}$,*

$$\left|\frac{\sum_{i=1}^{n} s_i^2(\langle \boldsymbol{x}_i, \boldsymbol{u}\rangle)^2}{\|\boldsymbol{s}\|_{\ell_2}^2} - \|\boldsymbol{u}\|_{\ell_2}^2\right| \leq \delta,$$

*holds with probability at least $1 - 6e^{-\frac{\delta^2}{1440}\|\boldsymbol{s}\|_{\ell_2}^2}$.*

## 6.2  Convergence proof (Proof of Theorem 3.1)

In this section we shall prove Theorem 3.1. Throughout, we use the shorthand $\mathcal{C}$ to denote the descent cone of $\mathcal{R}$ at $\boldsymbol{w}^*$, i.e. $\mathcal{C} = \mathcal{C}_{\mathcal{R}}(\boldsymbol{w}^*)$. We begin by analyzing the first iteration. Using $\boldsymbol{w}_0 = \boldsymbol{0}$ we have

$$\boldsymbol{w}_1 := \mathcal{P}_{\mathcal{K}}\left(\boldsymbol{w}_0 - \mu_0 \nabla \mathcal{L}(\boldsymbol{w}_0)\right) = \mathcal{P}_{\mathcal{K}}\left(\frac{2}{n}\sum_{i=1}^{n} y_i \boldsymbol{x}_i\right) = \mathcal{P}_{\mathcal{K}}\left(\frac{2}{n}\sum_{i=1}^{n} \text{ReLU}(\langle \boldsymbol{x}_i, \boldsymbol{w}^*\rangle)\boldsymbol{x}_i\right).$$

We use the argument of [21][Page 25, inequality (7.34)] which shows that

$$\|\boldsymbol{w}_1 - \boldsymbol{w}^*\|_{\ell_2} \le 2 \cdot \sup_{\boldsymbol{u} \in \mathcal{C} \cap \mathcal{B}^d} \boldsymbol{u}^T \left( \frac{2}{n} \sum_{i=1}^n \mathrm{ReLU}(\langle \boldsymbol{x}_i, \boldsymbol{w}^* \rangle) \boldsymbol{x}_i - \boldsymbol{w}^* \right). \tag{6.1}$$

Using $\mathrm{ReLU}(z) = \frac{z + |z|}{2}$ we have

$$\frac{2}{n} \sum_{i=1}^n \mathrm{ReLU}(\langle \boldsymbol{x}_i, \boldsymbol{w}^* \rangle) \langle \boldsymbol{x}_i, \boldsymbol{u} \rangle - \langle \boldsymbol{u}, \boldsymbol{w}^* \rangle = \boldsymbol{u}^T \left( \frac{1}{n} \boldsymbol{X}^T \boldsymbol{X} - \boldsymbol{I} \right) \boldsymbol{w}^* + \frac{1}{n} \sum_{i=1}^n |\langle \boldsymbol{x}_i, \boldsymbol{w}^* \rangle| \langle \boldsymbol{x}_i, \boldsymbol{u} \rangle. \tag{6.2}$$

We proceed by bounding the first term in the above equality. To this aim we decompose $\boldsymbol{u}$ in the direction parallel/perpendicular to that of $\boldsymbol{w}^*$ and arrive at

$$\boldsymbol{u}^T \left( \frac{1}{n} \boldsymbol{X}^T \boldsymbol{X} - \boldsymbol{I} \right) \boldsymbol{w}^* = \frac{(\boldsymbol{u}^T \boldsymbol{w}^*)}{\|\boldsymbol{w}^*\|_{\ell_2}^2} (\boldsymbol{w}^*)^T \left( \frac{1}{n} \boldsymbol{X}^T \boldsymbol{X} - \boldsymbol{I} \right) \boldsymbol{w}^* + \frac{1}{n} \left\langle \boldsymbol{X} \left( \boldsymbol{I} - \frac{\boldsymbol{w}^* (\boldsymbol{w}^*)^T}{\|\boldsymbol{w}^*\|_{\ell_2}^2} \right) \boldsymbol{u}, \boldsymbol{X} \boldsymbol{w}^* \right\rangle,$$

$$\sim (\boldsymbol{u}^T \boldsymbol{w}^*) \left( \frac{\|\boldsymbol{g}\|_{\ell_2}^2}{n} - 1 \right) + \frac{\|\boldsymbol{w}^*\|_{\ell_2}}{\sqrt{n}} \boldsymbol{a}^T \left( \boldsymbol{I} - \frac{\boldsymbol{w}^* (\boldsymbol{w}^*)^T}{\|\boldsymbol{w}^*\|_{\ell_2}^2} \right) \boldsymbol{u},$$

$$\le \|\boldsymbol{w}^*\|_{\ell_2} \left| \frac{\|\boldsymbol{g}\|_{\ell_2}^2}{n} - 1 \right| + \frac{\|\boldsymbol{w}^*\|_{\ell_2}}{\sqrt{n}} \sup_{\boldsymbol{u} \in \mathcal{C} \cap \mathcal{B}^d} \boldsymbol{a}^T \left( \boldsymbol{I} - \frac{\boldsymbol{w}^* (\boldsymbol{w}^*)^T}{\|\boldsymbol{w}^*\|_{\ell_2}^2} \right) \boldsymbol{u}, \tag{6.3}$$

with $\boldsymbol{g} \in \mathbb{R}^n$ and $\boldsymbol{a} \in \mathbb{R}^d$ are independent random Gaussian random vectors distributed as $\mathcal{N}(\boldsymbol{0}, \boldsymbol{I}_d)$ and $\mathcal{N}(\boldsymbol{0}, \boldsymbol{I}_n)$. By concentration of Chi-squared random variables

$$\left| \|\boldsymbol{g}\|_{\ell_2}^2 / n - 1 \right| \le \Delta, \tag{6.4}$$

holds with probability at least $1 - 2e^{-n \frac{\Delta^2}{8}}$. Also,

$$\frac{1}{\sqrt{n}} \boldsymbol{a}^T \left( \boldsymbol{I} - \frac{\boldsymbol{w}^* (\boldsymbol{w}^*)^T}{\|\boldsymbol{w}^*\|_{\ell_2}^2} \right) \boldsymbol{u} \le \frac{1}{\sqrt{n}} \left( \omega \left( \mathcal{C} \cap \mathcal{B}^d \right) + \eta \right), \tag{6.5}$$

holds with probability at least $1 - e^{-\frac{\eta^2}{2}}$. Plugging (6.4) with $\Delta = \frac{\delta}{6}$ and (6.5) with $\eta = \frac{\delta}{6} \sqrt{n}$ into (6.3), as long as $n \ge \frac{36}{\delta^2} \omega^2 \left( \mathcal{C} \cap \mathcal{B}^d \right)$, then

$$\sup_{\boldsymbol{u} \in \mathcal{C} \cap \mathcal{B}^d} \boldsymbol{u}^T \left( \frac{1}{n} \boldsymbol{X}^T \boldsymbol{X} - \boldsymbol{I} \right) \boldsymbol{w}^* \le \frac{\delta}{2} \|\boldsymbol{w}^*\|_{\ell_2}, \tag{6.6}$$

holds with probability at least $1 - 3e^{-n \frac{\delta^2}{288}}$.

We now focus on bounding the second term in (6.2). To this aim we decompose $\boldsymbol{u}$ in the direction parallel/perpendicular to that of $\boldsymbol{w}^*$ and arrive at

$$\left| \frac{1}{n} \sum_{i=1}^n |\langle \boldsymbol{x}_i, \boldsymbol{w}^* \rangle| \langle \boldsymbol{x}_i, \boldsymbol{u} \rangle \right| = \left| (\boldsymbol{u}^T \boldsymbol{w}^*) \frac{1}{n} \sum_{i=1}^n \frac{|\langle \boldsymbol{x}_i, \boldsymbol{w}^* \rangle| \langle \boldsymbol{x}_i, \boldsymbol{w}^* \rangle}{\|\boldsymbol{w}^*\|_{\ell_2}^2} + \frac{1}{n} \sum_{i=1}^n |\langle \boldsymbol{x}_i, \boldsymbol{w}^* \rangle| \langle \boldsymbol{x}_i, \boldsymbol{u}_\perp \rangle \right|,$$

$$\le \|\boldsymbol{w}^*\|_{\ell_2} \left| \frac{1}{n} \sum_{i=1}^n \frac{|\langle \boldsymbol{x}_i, \boldsymbol{w}^* \rangle| \langle \boldsymbol{x}_i, \boldsymbol{w}^* \rangle}{\|\boldsymbol{w}^*\|_{\ell_2}^2} \right| + \left| \frac{1}{n} \sum_{i=1}^n |\langle \boldsymbol{x}_i, \boldsymbol{w}^* \rangle| \langle \boldsymbol{x}_i, \boldsymbol{u}_\perp \rangle \right|. \tag{6.7}$$

with $\boldsymbol{u}_\perp = \left( \boldsymbol{I} - \frac{\boldsymbol{w}^* (\boldsymbol{w}^*)^T}{\|\boldsymbol{w}^*\|_{\ell_2}^2} \right) \boldsymbol{u}$. Now note that $\frac{|\langle \boldsymbol{x}_i, \boldsymbol{w}^* \rangle| \langle \boldsymbol{x}_i, \boldsymbol{w}^* \rangle}{\|\boldsymbol{w}^*\|_{\ell_2}^2}$ is sub-exponential and

$$\left\| \frac{|\langle \boldsymbol{x}_i, \boldsymbol{w}^* \rangle| \langle \boldsymbol{x}_i, \boldsymbol{w}^* \rangle}{\|\boldsymbol{w}^*\|_{\ell_2}^2} \right\|_{\psi_1} \le c,$$

with fixed numerical constant. Thus by Bernstein's type inequality ([24][Proposition 5.16])

$$\left| \frac{1}{n} \sum_{i=1}^n \frac{|\langle \boldsymbol{x}_i, \boldsymbol{w}^* \rangle| \langle \boldsymbol{x}_i, \boldsymbol{w}^* \rangle}{\|\boldsymbol{w}^*\|_{\ell_2}^2} \right| \le t, \tag{6.8}$$

holds with probability at least $1 - 2e^{-\gamma n \min(t^2, t)}$ with $\gamma$ a fixed numerical constant.. Also note that

$$\frac{1}{n}\sum_{i=1}^{n}|\langle \boldsymbol{x}_i, \boldsymbol{w}^* \rangle|\langle \boldsymbol{x}_i, \boldsymbol{u}_\perp \rangle \sim \sqrt{\frac{1}{n}\sum_{i=1}^{n}|\langle \boldsymbol{x}_i, \boldsymbol{w}^* \rangle|^2} \frac{1}{\sqrt{n}}\langle \boldsymbol{g}, \boldsymbol{u}_\perp \rangle.$$

Furthermore, $\frac{1}{n}\sum_{i=1}^{n}|\langle \boldsymbol{x}_i, \boldsymbol{w}^* \rangle|^2 \le (1+\Delta)\|\boldsymbol{w}^*\|_{\ell_2}^2$, holds with probability at least $1 - 2e^{-n\frac{\Delta^2}{8}}$ and

$$\sup_{\boldsymbol{u} \in \mathcal{C} \cap \mathbb{S}^{d-1}} |\langle \boldsymbol{g}, \boldsymbol{u}_\perp \rangle| \le (2\omega\left(\mathcal{C} \cap \mathbb{S}^{d-1}\right) + \eta),$$

holds with probability at least $1 - e^{-\frac{\eta^2}{2}}$. Combining the last two inequalities we conclude that

$$\left|\frac{1}{n}\sum_{i=1}^{n}|\langle \boldsymbol{x}_i, \boldsymbol{w}^* \rangle|\langle \boldsymbol{x}_i, \boldsymbol{u}_\perp \rangle\right| \le \sqrt{1+\Delta}\frac{(2\omega\left(\mathcal{C} \cap \mathbb{S}^{d-1}\right) + \eta)}{\sqrt{n}}\|\boldsymbol{w}^*\|_{\ell_2}, \tag{6.9}$$

holds with probability at least $1 - 2e^{-n\frac{\Delta^2}{8}} - e^{-\frac{\eta^2}{2}}$. Plugging (6.8) and (6.9) with $t = \frac{\delta}{6}$, $\Delta = 1$, and $\eta = \frac{\delta}{6\sqrt{2}}\sqrt{n}$ into (6.7)

$$\left|\frac{1}{n}\sum_{i=1}^{n}|\langle \boldsymbol{x}_i, \boldsymbol{w}^* \rangle|\langle \boldsymbol{x}_i, \boldsymbol{u} \rangle\right| \le \frac{\delta}{2}\|\boldsymbol{w}^*\|_{\ell_2}, \tag{6.10}$$

holds with probability at least $1 - 3e^{-\gamma n \delta^2} - 2e^{-\frac{n}{8}}$ as long as $n \ge 288\frac{\omega^2(\mathcal{C} \cap \mathbb{S}^{d-1})}{\delta^2}$. Thus pluggin (6.6) and (6.10) into (6.1) we conclude that for $\delta = 7/400$

$$\|\boldsymbol{w}_1 - \boldsymbol{w}^*\|_{\ell_2} \le 2 \cdot \sup_{\boldsymbol{u} \in \mathcal{C} \cap \mathcal{B}^d} \boldsymbol{u}^T\left(\frac{2}{n}\sum_{i=1}^{n}\text{ReLU}(\langle \boldsymbol{x}_i, \boldsymbol{w}^* \rangle)\boldsymbol{x}_i - \boldsymbol{w}^*\right) \le 2\delta\|\boldsymbol{w}^*\|_{\ell_2} \le \frac{7}{200}\|\boldsymbol{w}^*\|_{\ell_2},$$

holds with probability at least $1 - 8e^{-\gamma n}$ as long as $n \ge c\omega^2\left(\mathcal{C} \cap \mathbb{S}^{d-1}\right)$ for a fixed numerical constant $c$. To introduce our general convergence analysis we begin by defining

$$E(\epsilon) = \left\{\boldsymbol{w} \in \mathbb{R}^d : \mathcal{R}(\boldsymbol{w}) \le \mathcal{R}(\boldsymbol{w}^*), \ \|\boldsymbol{w} - \boldsymbol{w}^*\|_{\ell_2} \le \epsilon\|\boldsymbol{w}^*\|_{\ell_2}\right\} \quad \text{with} \quad \epsilon = \frac{7}{200}.$$

To prove Theorem 3.1 we use [21][Page 25, inequality (7.34)] which shows that if we apply the projected gradient descent update $\boldsymbol{w}_{\tau+1} = \mathcal{P}_\mathcal{K}\left(\boldsymbol{w}_\tau - \nabla\mathcal{L}(\boldsymbol{w}_\tau)\right)$, the error $\boldsymbol{h}_\tau = \boldsymbol{w}_\tau - \boldsymbol{w}^*$ obeys

$$\|\boldsymbol{h}_{\tau+1}\|_{\ell_2} = \|\boldsymbol{w}_{\tau+1} - \boldsymbol{w}^*\|_{\ell_2} \le 2 \cdot \sup_{\boldsymbol{u} \in \mathcal{C} \cap \mathcal{B}^n} \boldsymbol{u}^*\left(\boldsymbol{h}_\tau - \nabla\mathcal{L}(\boldsymbol{w}_\tau)\right). \tag{6.11}$$

To complete the convergence analysis it is then sufficient to prove

$$\sup_{\boldsymbol{u} \in \mathcal{C} \cap \mathcal{B}^n} \boldsymbol{u}^*\left(\boldsymbol{h}_\tau - \nabla\mathcal{L}(\boldsymbol{w}_\tau)\right) \le \frac{1}{4}\|\boldsymbol{h}_\tau\|_{\ell_2} = \frac{1}{4}\|\boldsymbol{w}_\tau - \boldsymbol{w}^*\|_{\ell_2}. \tag{6.12}$$

We will instead prove that the following stronger result holds for all $\boldsymbol{u} \in \mathcal{C} \cap \mathcal{B}^n$ and $\boldsymbol{w} \in E(\epsilon)$

$$\boldsymbol{u}^*\left(\boldsymbol{w} - \boldsymbol{w}^* - \nabla\mathcal{L}(\boldsymbol{w})\right) \le \frac{1}{4}\|\boldsymbol{w} - \boldsymbol{w}^*\|_{\ell_2}. \tag{6.13}$$

The equation (6.13) above implies (6.12) which when combined with (6.11) proves the convergence result of the Theorem (specifically equation (3.5)). The rest of this section is dedicated to proving (6.13). To this aim note that $\text{ReLU}(\langle \boldsymbol{x}_i, \boldsymbol{w} \rangle) = \frac{\langle \boldsymbol{x}_i, \boldsymbol{w} \rangle + |\langle \boldsymbol{x}_i, \boldsymbol{w} \rangle|}{2}$. Thus (see the extended version of this paper [20] for more detailed derivation of the identity below)

$$\langle \nabla\mathcal{L}(\boldsymbol{w}), \boldsymbol{u} \rangle = \frac{1}{n}\sum_{i=1}^{n}\langle \boldsymbol{x}_i, \boldsymbol{w} - \boldsymbol{w}^* \rangle\langle \boldsymbol{x}_i, \boldsymbol{u} \rangle + \frac{1}{n}\sum_{i=1}^{n}\text{sgn}(\langle \boldsymbol{x}_i, \boldsymbol{w}^* \rangle)\langle \boldsymbol{x}_i, \boldsymbol{w} - \boldsymbol{w}^* \rangle\langle \boldsymbol{x}_i, \boldsymbol{u} \rangle$$

$$+ \frac{1}{n}\sum_{i=1}^{n}\left(\text{sgn}(\langle \boldsymbol{x}_i, \boldsymbol{w} \rangle) - \text{sgn}(\langle \boldsymbol{x}_i, \boldsymbol{w}^* \rangle)\right)\langle \boldsymbol{x}_i, \boldsymbol{w} - \boldsymbol{w}^* \rangle\langle \boldsymbol{x}_i, \boldsymbol{u} \rangle$$

$$+ \frac{1}{2n}\sum_{i=1}^{n}\left(1 - \text{sgn}(\langle \boldsymbol{x}_i, \boldsymbol{w}^* \rangle)\right)\left(\text{sgn}(\langle \boldsymbol{x}_i, \boldsymbol{w}^* \rangle) - \text{sgn}(\langle \boldsymbol{x}_i, \boldsymbol{w} \rangle)\right)|\langle \boldsymbol{x}_i, \boldsymbol{w}^* \rangle|\langle \boldsymbol{x}_i, \boldsymbol{u} \rangle$$

Now defining $\boldsymbol{h} = \boldsymbol{w} - \boldsymbol{w}^*$ we conclude that $\langle \boldsymbol{u}, \boldsymbol{w} - \boldsymbol{w}^* - \nabla\mathcal{L}(\boldsymbol{w})\rangle = \langle \boldsymbol{u}, \boldsymbol{h} - \nabla\mathcal{L}(\boldsymbol{w})\rangle$ is equal to

$$
\begin{aligned}
\langle \boldsymbol{u}, \boldsymbol{h} - \nabla\mathcal{L}(\boldsymbol{w})\rangle = &\boldsymbol{u}^T\left(\boldsymbol{I} - \frac{1}{n}\boldsymbol{X}\boldsymbol{X}^T\right)\boldsymbol{h} - \frac{1}{n}\sum_{i=1}^{n}\mathrm{sgn}(\langle \boldsymbol{x}_i, \boldsymbol{w}^*\rangle)\langle \boldsymbol{x}_i, \boldsymbol{h}\rangle\langle \boldsymbol{x}_i, \boldsymbol{u}\rangle, \\
&+ \frac{\langle \boldsymbol{h}, \boldsymbol{w}^*\rangle}{\|\boldsymbol{w}^*\|_{\ell_2}^2}\frac{1}{n}\sum_{i=1}^{n}\left(1 - \mathrm{sgn}(\langle \boldsymbol{x}_i, \boldsymbol{w}\rangle)\mathrm{sgn}(\langle \boldsymbol{x}_i, \boldsymbol{w}^*\rangle)\right)\mathrm{sgn}(\langle \boldsymbol{x}_i, \boldsymbol{w}^*\rangle)\langle \boldsymbol{x}_i, \boldsymbol{h}\rangle\langle \boldsymbol{x}_i, \boldsymbol{u}\rangle, \\
&+ \frac{\mathrm{sgn}(\langle \boldsymbol{x}_i, \boldsymbol{w}\rangle)}{2n}\sum_{i=1}^{n}\left(1 - \mathrm{sgn}(\langle \boldsymbol{x}_i, \boldsymbol{w}^*\rangle)\right)\left(1 - \mathrm{sgn}(\langle \boldsymbol{x}_i, \boldsymbol{w}\rangle)\mathrm{sgn}(\langle \boldsymbol{x}_i, \boldsymbol{w}^*\rangle)\right) \\
&\qquad\qquad |\langle \boldsymbol{x}_i, \boldsymbol{w}^*\rangle|\,\langle \boldsymbol{x}_i, \boldsymbol{u}\rangle.
\end{aligned}
$$

Now define $\boldsymbol{h}_\perp = \boldsymbol{h} - (\boldsymbol{h}^T\boldsymbol{w}^*)/(\|\boldsymbol{w}^*\|_{\ell_2}^2)\boldsymbol{w}^*$. Using this we can rewrite the previous expression in the form (see the proof in the extended version of this paper [20] for more detailed derivation)

$$
\begin{aligned}
\langle \boldsymbol{u}, \boldsymbol{w} - \boldsymbol{w}^* - \nabla\mathcal{L}(\boldsymbol{w})\rangle = &\boldsymbol{u}^T\left(\boldsymbol{I} - \frac{1}{n}\boldsymbol{X}\boldsymbol{X}^T\right)\boldsymbol{h} - \frac{1}{n}\sum_{i=1}^{n}\mathrm{sgn}(\langle \boldsymbol{x}_i, \boldsymbol{w}^*\rangle)\langle \boldsymbol{x}_i, \boldsymbol{h}\rangle\langle \boldsymbol{x}_i, \boldsymbol{u}\rangle, \\
&+ \frac{1}{n}\sum_{i=1}^{n}\left(1 - \mathrm{sgn}(\langle \boldsymbol{x}_i, \boldsymbol{w}\rangle)\mathrm{sgn}(\langle \boldsymbol{x}_i, \boldsymbol{w}^*\rangle)\right)\mathrm{sgn}(\langle \boldsymbol{x}_i, \boldsymbol{w}^*\rangle)\langle \boldsymbol{x}_i, \boldsymbol{h}_\perp\rangle\langle \boldsymbol{x}_i, \boldsymbol{u}\rangle, \\
&+ \frac{1}{n}\sum_{i=1}^{n}\left[\frac{\mathrm{sgn}(\langle \boldsymbol{x}_i, \boldsymbol{w}\rangle)}{2}\left(1 - \mathrm{sgn}(\langle \boldsymbol{x}_i, \boldsymbol{w}^*\rangle)\right) + \frac{\langle \boldsymbol{h}, \boldsymbol{w}^*\rangle}{\|\boldsymbol{w}^*\|_{\ell_2}^2}\right] \\
&\qquad\qquad \left(1 - \mathrm{sgn}(\langle \boldsymbol{x}_i, \boldsymbol{w}\rangle)\mathrm{sgn}(\langle \boldsymbol{x}_i, \boldsymbol{w}^*\rangle)\right)|\langle \boldsymbol{x}_i, \boldsymbol{w}^*\rangle|\,\langle \boldsymbol{x}_i, \boldsymbol{u}\rangle \qquad (6.14)
\end{aligned}
$$

We now proceed by stating bounds on each of the four terms in (6.14). The detailed derivation of these bounds appear in the the extended version of this paper [20].

**Lemma 6.5** *Assume the setup of Theorem 3.1. Then as long as $n \ge cn_0$, we have*

$$
\boldsymbol{u}^*\left(\boldsymbol{I} - \frac{1}{n}\boldsymbol{X}^*\boldsymbol{X}\right)\boldsymbol{h} \le \delta\,\|\boldsymbol{h}\|_{\ell_2}, \qquad (6.15)
$$

$$
-\frac{1}{n}\sum_{i=1}^{n} sgn(\langle \boldsymbol{x}_i, \boldsymbol{w}^*\rangle)\langle \boldsymbol{x}_i, \boldsymbol{h}\rangle\langle \boldsymbol{x}_i, \boldsymbol{u}\rangle \le \delta\,\|\boldsymbol{h}\|_{\ell_2}, \qquad (6.16)
$$

$$
\frac{1}{n}\sum_{i=1}^{n}\left(1 - sgn(\langle \boldsymbol{x}_i, \boldsymbol{w}\rangle)sgn(\langle \boldsymbol{x}_i, \boldsymbol{w}^*\rangle)\right)sgn(\langle \boldsymbol{x}_i, \boldsymbol{w}^*\rangle)\langle \boldsymbol{x}_i, \boldsymbol{h}_\perp\rangle\langle \boldsymbol{x}_i, \boldsymbol{u}\rangle \le 2\sqrt{1+\delta}\left(\delta + \sqrt{\frac{21}{20}}\epsilon\right)\|\boldsymbol{h}\|_{\ell_2}, \qquad (6.17)
$$

$$
\begin{aligned}
&\frac{1}{n}\sum_{i=1}^{n}\left[\frac{sgn(\langle \boldsymbol{x}_i, \boldsymbol{w}\rangle)}{2}\left(1 - sgn(\langle \boldsymbol{x}_i, \boldsymbol{w}^*\rangle)\right) + \frac{\langle \boldsymbol{h}, \boldsymbol{w}^*\rangle}{\|\boldsymbol{w}^*\|_{\ell_2}^2}\right] \\
&\left(1 - sgn(\langle \boldsymbol{x}_i, \boldsymbol{w}\rangle)sgn(\langle \boldsymbol{x}_i, \boldsymbol{w}^*\rangle)\right)|\langle \boldsymbol{x}_i, \boldsymbol{w}^*\rangle|\,\langle \boldsymbol{x}_i, \boldsymbol{u}\rangle \le \frac{4\sqrt{1+\delta}}{(1-\epsilon)^2}\left(\delta + \sqrt{\frac{21}{20}}\epsilon\right)\|\boldsymbol{h}\|_{\ell_2}, \qquad (6.18)
\end{aligned}
$$

*holds for all $\boldsymbol{u} \in \mathcal{C} \cap \mathbb{S}^{d-1}$ and $\boldsymbol{w} \in E(\epsilon)$ with probability at least $1 - 9e^{-\gamma n}$.*

Combining (6.15), (6.16), (6.17), and (6.18) we conclude that

$$
\langle \boldsymbol{u}, \boldsymbol{w} - \boldsymbol{w}^* - \nabla\mathcal{L}(\boldsymbol{w})\rangle \le 2\left(\delta + \sqrt{1+\delta}\left(1 + \frac{2}{(1-\epsilon)^2}\right)\left(\delta + \sqrt{\frac{21}{20}}\epsilon\right)\right)\|\boldsymbol{w} - \boldsymbol{w}^*\|_{\ell_2},
$$

holds for all $\boldsymbol{u} \in \mathcal{C} \cap \mathbb{S}^{d-1}$ and $\boldsymbol{w} \in E(\epsilon)$ with probability at least $1 - 16e^{-\gamma\delta^2 n} - (n+10)e^{-\gamma n}$. Using this inequality with $\delta = 10^{-4}$ and $\epsilon = 7/200$ we conclude that $\langle \boldsymbol{u}, \boldsymbol{w} - \boldsymbol{w}^* - \nabla\mathcal{L}(\boldsymbol{w})\rangle \le \frac{1}{4}\|\boldsymbol{w} - \boldsymbol{w}^*\|_{\ell_2}$, holds for all $\boldsymbol{u} \in \mathcal{C} \cap \mathbb{S}^{d-1}$ and $\boldsymbol{w} \in E(\epsilon)$ with high probability.

# Acknowledgements

This work was done in part while the author was visiting the Simon's Institute for the Theory of Computing. M.S. would like to thank Adam Klivans and Matus Telgarsky for discussions related to [6] and the Isotron algorithm.

## Footnotes

[1]We would like to note that $n_0$ only approximately characterizes the minimum number of samples required. A more precise characterization is $\phi^{-1}(\omega^2(\mathcal{C}_{\mathcal{R}}(\boldsymbol{w}^*) \cap \mathcal{B}^d)) \approx \omega^2(\mathcal{C}_{\mathcal{R}}(\boldsymbol{w}^*) \cap \mathcal{B}^d)$ where $\phi(t) = \sqrt{2} \frac{\Gamma(\frac{t+1}{2})}{\Gamma(\frac{t}{2})} \approx \sqrt{t}$. However, since our results have unspecified constants we avoid this more accurate characterization.

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
