[Reviews · NeurIPS 2017]

Reviewer 1



### Summary The paper proves that projected gradient descent in a single-layer ReLU network converges to the global minimum linearly: ||w_t-w^*||/||w_0-w^*||=(1/2)^t with high probability. The analysis assumes the input samples are iid Gaussian and the output is realizable, i.e. the target value y_k for input x_k is constructed via y_k=ReLU(w^*.x_k). The paper studies a regression setting and uses least squares as the loss function. ### Pros and Cons To the best of my knowledge, the result is novel; I am not aware of any proof of "convergence to global optimum" for the nonconvex loss resulted by fitting to a ReLu function via least squares. On top of that, the analysis considers a nonconvex regularization term with little assumptions on it. Of course, there still a clear gap between the results of this paper and models used in practice: notably, relaxing the Gaussian iid input assumption, and extending to multilayer setup. However, this result may become a stepping stone toward that goal. ### Proofs I followed the building blocks of the proof at the coarse level, and the logic makes sense. However, I did not check at a very detailed level. I was wondering if the authors could include a simple simulation to confirm the correctness of the result? Given that the data is very simple (iid Gaussian) and the network is a single layer ReLU, using a very simple regularization scheme, it seems to take a few lines of Matlab to verify the empirical convergence rate and compare that with the result. In fact, I wonder why this is not attempted by the authors already? ### Regularization The authors state that the result holds true regardless of regularization R being convex or nonconvex. What if R is such that it creates two far apart and disjoint regions, one containing the origin (initialization) and the other containing w^*. Wouldn't this create a problem for projected gradient descent initialized by w=0 to reach w^*? ### Minor Comments 1. There are typos, please proof read. 2. What is the meaning of A in L_A (Eq 5.12)? 3. What is the value of the constant gamma in line 180, or otherwise how is it related to other constants or quantities in the paper?

Reviewer 2



Review: Learning ReLUs via Gradient Descent This paper study the problem of learning the parameters of ReLUs by using the gradient descent algorithm. The authors have shown that under certain condition of the total observed samples, performing gradient descent is able to accurately estimate the true parameter w^* for a 1-layer ReLu network, with high probability. Generally the paper is well written as easy to follow. I have only a few minor comments. 1) The random vector g in line 168 of the paper maybe different from the g defined in the equation about (5.8), please check and comment. 2) For the last two relations in page 6 (one line 181 and 182), please cite the related sources. 3) Please comment on which lemma is used to derive Eq. (5.5). 4) Last line in page 8, the term "+ \frac{sgn…}{2n}" should be included in the summation sign. 5) I am not exactly sure how the first equality of (5.14) is derived. It appears that a term "< h^T, w^* >/||w^*||^2" is missing? Please double check.

Reviewer 3



This paper analyzes when gradient descent algorithm can be used to learn a single neuron with ReLU activation function. The main result is when the input is generated from Gaussian distribution, and the output is exactly from a ReLU unit, gradient descent algorithm has geometric convergence starting from 0 with fairly tight sample complexity (the sample complexity depend on the structure of the regularizer). The techniques of the result are quite interesting, and different from the traditional approaches for analyzing gradient descent. The approach is most similar to a recent analysis of phase retrieval problem [15]. Pros: - The paper provides a new analysis for learning ReLU unit, and the algorithm is simply gradient descent. - The sample complexity seems fairly tight in many settings and especially works in high dimensions (when number of samples is smaller) - Can handle some non-convex regularizers. Cons: - The analysis relies fairly heavily on the Gaussian distribution and realizable assumption. One conjecture is whether the objective function is actually convex when taking expectation over Gaussian? It might be useful to discuss a bit more about that. - To handle non-convex projection, the analysis seems fairly sensitive in the step size. In particular, unlike traditional analysis it seems like the proof will not work with a small step size (because the non-convex projection might push the point back). Overall this is a well-written paper with an interesting result.